# Magnesium Sulfate, Rosuvastatin, Sildenafil and Their Combination in Chronic Hypoxia-Induced Pulmonary Hypertension in Male Rats

**DOI:** 10.3390/life14091193

**Published:** 2024-09-20

**Authors:** Silvana-Elena Hojda, Irina Camelia Chis, Simona Clichici

**Affiliations:** Department of Physiology, “Iuliu Hatieganu” University of Medicine and Pharmacy, Number 1–3, Clinicilor Street, RO-400023 Cluj-Napoca, Romania; sclichici@umfcluj.ro

**Keywords:** animal model, chronic hypoxia-induced pulmonary hypertension, magnesium sulfate, rosuvastatin, sildenafil

## Abstract

Previous experimental findings have led to considerable interest in the beneficial effects on pulmonary hypertension (PH) produced by sildenafil and in the pleiotropic effects of rosuvastatin and their positive role in the process of pulmonary angiogenesis. However, magnesium sulfate, the most abundant intracellular cation, is essential in vascular endothelial functionality due to its anti-inflammatory and vasodilatory effects. Therefore, the present study aims to assess these treatment regimens and how they could potentially provide some additional benefits in PH therapy. Fourteen days after chronic-hypoxia PH was induced, rosuvastatin, sildenafil and magnesium sulfate were administered for an additional fourteen days to male Wistar rats. The Fulton Index, right ventricle (RV) anterior wall thickness, RV internal diameter and pulmonary arterial (PA) acceleration time/ejection time were evaluated, and another four biochemical parameters were calculated: brain natriuretic peptide, vascular endothelial growth factor, nitric oxide metabolites and endothelin 1. The present study demonstrates that sildenafil and rosuvastatin have modest effects in reducing RV hypertrophy and RV systolic pressure. The drug combination of sildenafil + rosuvastatin + magnesium sulfate recorded statistically very highly significant results on all parameters; through their positive synergistic effects on vascular endothelial function, oxidative stress and pathological RV remodeling, they attenuated PH in the chronic hypoxia pulmonary hypertension (CHPH) rat model.

## 1. Introduction

Pulmonary hypertension (PH) is a severe and heterogeneous vascular pathology characterized by an increase in mean pulmonary artery (PA) pressure above 20 mmHg [1]. Based on its etiology, PH is classified into five major groups by the World Health Organization. Group 1 includes PA hypertension (PAH), which comprises several subcategories. PAH is characterized by excessive arterial vasoconstriction and vascular cell proliferation accompanied by pulmonary vascular resistance (PVR) > 3 wood units (WUs) in the absence of other causes of precapillary PH [2]. Group 2 is a secondary characteristic of left-sided heart disease, and group 3 occurs due to lung disease and/or alveolar hypoxia, such as chronic obstructive pulmonary disease (COPD) and/or emphysema, interstitial lung disease, pulmonary fibrosis and hypoventilation syndromes. PVR > 5 WUs is the threshold where prognosis worsens. These represent the most common forms of PH. Group 4 occurs in relation to chronic thromboembolic disease, and lastly, group 5 covers several forms of PH with multifactorial mechanisms [2].

Despite all the advances in medical science, PH remains a life-threatening disease with a guarded prognosis. The main therapeutic objectives for this pathology are an improvement in vascular dysfunction, a reduction in inflammation, the amelioration of oxidative stress, the inhibition of pathological remodeling of the right ventricle (RV) and the preservation of RV function as much as possible [3].

Animal models are therefore indispensable tools in research studies for identifying new disease mechanisms or therapeutic targets. Over time, more and more animal models of PH have been developed to try to replicate human PH evolution, but none have fulfilled all the necessary characteristics of human PH.

The chronic hypoxia-induced PH (CHPH) animal model is one of the most classic experimental models of PH induction and, therefore, the most commonly used [3]. It is an experimental model with good predictability and repeatability, simple implementation and a low cost. Specifically, rats and mice are the most used animals.

After 3–4 weeks of exposure to chronic hypoxia at 10% inspired O_2_, this experimental model induces moderate pulmonary hypertension through two pathophysiologic mechanisms: increased vascular tone and vascular remodeling [4].

During exposure to CH, arteriolar vasospasm occurs in the pulmonary vascular bed, followed by endothelial cell dysfunction and apoptosis, inflammation and oxidative stress, which lead to the release of vasoactive factors and to a decrease in the endothelium’s capacity to produce vasodilator factors [5].

Echocardiography, a non-invasive method for monitoring disease progression and evaluating treatment effectiveness, is considered a gold standard technique in PH diagnosis. Several studies in the literature report the sensitivity and specificity of certain echocardiographic parameters, validated by invasive hemodynamic measurements [6,7]. An anterior RV wall thickness > 1.03 mm predicts a Fulton Index > 0.34 with 75% sensitivity and 76% specificity in different rat models of PH [6], and a pulmonary arterial acceleration time/ejection time ratio (PAAT/PAET) < 0.25 predicts PH with a sensitivity of 77% and specificity of 80%, correlated with an RV systolic pressure > 35 mmHg on hemodynamic measurements [6].

Hypoxia-induced pulmonary vascular remodeling is characterized by the development of smooth muscle-like cells involved in the medial hypertrophy and muscularization of pulmonary arterioles, cardiac fibrosis and pathological RV remodeling [8].

Biochemical parameters are an essential non-invasive approach to PH diagnosis and staging. One or more corresponding biomarkers of these pathophysiological mechanisms at the endothelial and vascular levels have been described in the literature [9]. Some of them are also evaluated in the present article.

Endothelin-1 (ET-1) is a potent vasoconstrictor factor and an important prognostic marker. Endothelin receptor antagonists have proven their efficacy in PAH, but their use in group 3 of PH has not yielded satisfactory results for this patient category [2].

Vascular endothelial growth factor (VEGF) is a circulating angiogenic modulatory factor, a marker of endothelial cell dysfunction implicated in vascular remodeling and very abundant in the lungs, with increased plasma concentration values in patients with PH [9].

Nitric oxide (NO) is an essential vasodilatory factor in maintaining normal vascular tone. Secondary to hypoxemia, an alteration in the NO pathway occurs in the pulmonary endothelium, which favors increased vascular tone and pathological remodeling in the pulmonary arterioles [10]. The biotransformation of NO into nitrite/nitrate metabolites takes place in the body through various pathways [11]. The increased amount of NO metabolites from various pathologies can be explained by the hyper-reactivity of NOS enzymes secondary to pro-inflammation.

Brain natriuretic peptides (BNPs) are molecules released by cardiac myocytes in response to RV pressure and volume overload. BNPs stimulate diuresis and natriuresis and have veno- and arteriolar dilator effects [2,9]. They are used in practice together with the N-terminal fragment of BNPs for the screening, diagnosis and monitoring of patients with chronic heart failure [12].

In what concerns PH, current treatments with endothelin receptor antagonists, phosphodiesterase 5 (PDE5) inhibitors (such as sildenafil) or prostacyclin analogs may, to some extent, improve the symptoms of PAH group 1 patients. At present, according to current guidelines [2], the use of PAH-specific medication in PH group 3 has limitations and clear discordances. In terms of specific therapy with sildenafil, clinical trials with a small number of patients have demonstrated that PDE5 has significantly improved PVR, the 6 min walk distance test and the quality of life in patients with COPD and severe PH [13]. However, in the absence of large-scale randomized trials, this evidence is still insufficient to support the use of sildenafil for this patient category [2,14]. This patient category often benefits from treatment optimization for the underlying pulmonary pathology (supplementary oxygen, non-invasive ventilation and pulmonary rehabilitation programs).

For this reason, experimental CHPH models are useful in the evaluation of physiologic and pathophysiologic processes of human PH associated with chronic lung diseases and for testing various innovative pharmacologic treatments [3,4].

Moreover, the efficacy and safety of statin treatment have been verified over time in COPD patients. Statins, such as hydroxy-methyl-glutaryl-coenzyme (HMG-CoA) reductase inhibitors, are characterized by beneficial pleiotropic effects: antiproliferative; antioxidant; anti-inflammatory; and finally, hypolipidemic effects. Statins may have a protective effect against group 3 of PH but are time- and dose-dependent [15]. Experimental studies have demonstrated that statins offer possible protection against pulmonary vascular remodeling, PH development and RV hypertrophy in animal models of PH [16].

The additive effects of the sildenafil + rosuvastatin combination were studied in the PH rat model. This treatment regimen showed a modest but important effect in improving the pathological vascular remodeling of pulmonary arterioles, implicitly decreasing PVR, with beneficial effects on RV functionality. Hepatotoxicity due to statins can be prevented with the use of low drug doses [17].

Magnesium, the most abundant intracellular cation, is involved in multiple cellular processes and is an essential element for normal vasoreactivity of the pulmonary endothelium. Increasing magnesium concentration improves vasodilatation by releasing vasodilator elements that exhibit anti-inflammatory and antioxidant properties [18]. Magnesium sulfate is a natural calcium antagonist. It can upregulate intracellular Ca^2+^ influx through its mimetic Ca^2+^ channel-blocking effect, stimulate the production of vasodilator prostacyclin and nitric oxide, and modify vascular responses to vasoactive agonists. These biochemical reactions control vascular contraction, dilation, growth, apoptosis, and inflammation. Magnesium also suppresses the inhibitory activity of circulating Na + K + ATP ase, which reduces vascular tone [19]. By activating NO synthase (NOS), magnesium increases the release of NO endothelium, a highly reactive molecule involved in various complex physiological processes. NO is a crucial modulator of vascular homeostasis, maintains vascular tone, and has antiplatelet and antithrombotic properties. Endothelial cells release a constitutive form of NO synthase (eNOS), which generates NO under physiologic conditions. High levels of extracellular Mg stimulate NO production by increasing eNOS levels, whereas no data are available on the effects of low Mg on NO synthesis in human endothelial cells [20].

Dysfunctions in magnesium transport systems may predispose patients to cardiovascular disease pathologies. Magnesium efflux occurs via Na++-dependent and Na++-independent pathways. Magnesium influx is controlled by TRPM7 (transient receptor potential melastatin), a non-selective activator of the Ca^2+^ channel. Thus, TRPM7 may influence intracellular magnesium levels through alterations in the efflux and influx. This magnesium in red blood cells is a more accurate reflection of total body magnesium stores [19].

It is most commonly used in the treatment of eclampsia and cardiac arrhythmias [21,22]. Short-term magnesium administration reduced pulmonary vasoconstriction in premature neonates and newborns with PH [23], as well as in PH rat models [24,25].

Magnesium deficiency may be a pro-inflammatory trigger and may favor exacerbations in COPD patients through bronchoconstriction. A recent clinical trial has demonstrated that oral magnesium supplementation in COPD patients may play a bronchodilator role by improving the response to bronchial musculature but without influencing the quality of life or physical performance of these patients [26].

In this study, we created an experimental CHPH model. Subsequently, by administering sildenafil, rosuvastatin, magnesium sulfate and a combination of these three, we evaluated animals anatomically (through RV hypertrophy parameters), echocardiographically and biologically.

## 2. Materials and Methods

### 2.1. Animal Care and Ethical Approval

Forty-two male Wistar rats, body mass 200–250 g, fed 1.0% methylcellulose, 1 mg kg^−1^, intragastrically (i.g.) daily and with free access to water were used in this experiment. The experimental protocol was performed in the Department of Physiology of the Faculty of Medicine in Cluj-Napoca, Romania. Animal housing, medication administration and the entire experiment protocol were performed according to the International Animal Research Guidelines and were approved by the Animal Care and Use Committee of both the University “Iuliu Hațieganu” and Veterinary Health Authority of Cluj-Napoca, Romania (no. 229/12.08.2020; 260/22.07.2020), and respecting, also, Directive 86/609/EEC. The animals came from the Center for Experimental Medicine and Practical Skills of the University of Medicine and Pharmacy “Iuliu Hațieganu”, Cluj-Napoca, Romania. Every effort was made to reduce animal suffering and to use as few animals as possible while still providing statistically significant results.

### 2.2. Experimental Design

The rats were housed in an automated hypobaric chamber for 4 weeks and exposed to chronic hypoxia at 10% inspired O_2_ at a pressure corresponding to 5500 m altitude (as shown in Figure 1). The 42 rats were divided into 6 equal groups. The animals were kept in special cages, 7 animals per cage. The control group (C) was kept at normal atmospheric pressure. The animals in groups I, II, II, IV and V were placed in a hypobaric chamber for 4 weeks. Two weeks after exposure to hypobaric hypoxia, groups II, III, IV and V were administered the daily treatment and continued to be kept under hypoxic conditions. Since CHPH is partially reversible upon atmospheric pressure normalization, we preferred to keep the rats under hypoxic conditions. All drugs, in the dosages detailed below, were administered via intragastric gavage and suspended in 1.0% methylcellulose.

### 2.3. Animal Treatment

Group I consisted of 7 rats subjected only to 10% hypoxia. Group II received sildenafil, 25 mg/kg, i.g. daily for 14 days. Group III received sildenafil, 25 mg/kg, i.g. + rosuvastatin, 10 mg/kg, i.g. daily for 14 days. Group IV received sildenafil, 25 mg/kg, i.g. + magnesium sulfate 10% solution, 10 mL/kg, i.g. daily for 14 days. Finally, group V received a combination of the three: sildenafil, 25 mg/kg, i.g. + rosuvastatin, 10 mg/kg, i.g. + magnesium sulfate 10% solution, 10 mL/kg, i.g. daily for 14 days.

After 4 weeks from inclusion in the study, all 42 rats survived and were subjected to further investigation. Adverse reactions from the medication were negligible, with no life-threatening effects.

### 2.4. Echocardiographic Measurements

Then, 24 h after the last treatment dose, the animals were sedated with pentobarbital sodium (30 mg/kg) via intraperitoneal injection. The animals were placed in supine position, and echocardiographic measurements were performed using an Ultrasonix version 6.0.7 portable ultrasound scanner and a 20 Mhz MicroScan solid-state transducer (Visual Sonics, Toronto, ON, Canada). Additional boluses of pentobarbital sodium were administered as needed in order to maintain sedation and a heart rate of approximately 300 beats per minute.

Various markers of pulmonary hemodynamics and RV hypertrophy were measured in the rat model according to international recommendations for echocardiographic PH assessment [7,8]: from the parasternal long-axis view using B-mode echocardiography, RV anterior wall thickness and RV internal diameter measured in diastole; from the parasternal short-axis view of the modified section, the pulmonary artery (PA) diameter; and the pulsed-wave Doppler sample volume placed in the center of PA for pulmonary artery acceleration time (PAAT) and pulmonary artery ejection time (PAET).

### 2.5. Measurement of Body and Organ Mass

At the end of the study, the animals were euthanized with an overdose of pentobarbital sodium (100 mg/kg) administered via intraperitoneal injection. The animals were then weighed using a veterinary scale. The heart and lungs were removed through a medial thoracic incision. Immediately after removal, the hearts were blotted dry and weighed. The right ventricle (RV) was then separated from the left ventricle and septum (LV + S) and weighed immediately. The Fulton Index (the right ventricle mass index (RVMI) = RV/LV + S) was calculated as an index for RV hypertrophy.

### 2.6. Measurements of ET-1, BNP, VEGF and NO Metabolite Levels

The levels of these biomarkers were measured with commercially available ELISA kits for rats (Sigma-Aldrich, Bucharest, Romania) in PA and RV tissue samples.

### 2.7. Statistical Analysis

Descriptive statistics were calculated. Normal distribution was tested with the Shapiro–Wilk test, and variance with the F-test. The two groups were compared using Student’s *t*-test or Mann–Whitney U tests. The statistical significance thresholds were α = 0.05 (5%), α = 0.01 (1%) and α = 0.001 (0.1%), as follows: *p* < 0.05—statistically significant difference; 0.001 < *p* < 0.01—statistically highly significant difference; and *p* < 0.001—statistically very highly significant difference.

Statistical processing was performed with the software StatsDirect v.2.7.2 and the Excel spreadsheet editor (Microsoft Office 2019 package).

## 3. Results

### 3.1. CHPH in Rats

In order to evaluate the success of PH development in the experimental model, we compared the measurements of body and organ mass, the echocardiographic measurements and the biochemical parameters of group I with those of the control group. The results were highly significant (*p* < 0.001) between groups C and I in terms of RV weight increase. The highest RV weight was recorded in group I after 4 weeks of exposure to 10% hypoxia. Moreover, the Fulton Index, considered a parameter of RV hypertrophy, had highly significant results in group I compared with group C, as shown in Figure 2. Echocardiographically, the increase in RV anterior wall thickness was statistically significant (*p* < 0.001) (Figure 3). The indicated cutoff value of RV anterior wall thickness > 1.03 predicts a Fulton Index > 0.34 with 75% sensitivity and 76% specificity. A statistically significant RV dilatation was also observed in group I compared with group C (Figure 4). Two indirect parameters of increased pulmonary vascular resistance (PVR) were also compared: PAAT and PAET. The PAAT/PAET ratio had statistically highly significant results (*p* < 0.001) in group I compared with group C, with values of this ratio < 0.25 (Figure 5). In addition, the BNP value in the PA tissue and RV tissue samples in group I was greatly increased, statistically very highly significant compared with group C (*p* < 0.001) (Figure 6 and Figure 7). The increases in VEGF and NO metabolite values in both evaluated tissue samples were also statistically significant (*p* < 0.05) (Figure 8 and Figure 9). All the results described above support the fact that the experimental model of hypoxia-induced PH is successful in modeling pressure and volume overload at RV and PA, RV hypertrophy, myocardial stress and PH.

### 3.2. Measurement of Body and Organ Mass after Treatments

In the statistical analysis of the Fulton Index values, we observed statistically very highly significant differences between groups II and V and groups III and V. In the absence of medication, the Fulton Index registers a ratio of approximately 0.34. However, under treatment, there is a progressive reduction, as shown in Figure 2. The administration of sildenafil, 25 mg/kg, i.g. + rosuvastatin, 10 mg/kg, i.g. + magnesium sulfate 10% solution, 10 mL/kg, i.g. in group V compared with sildenafil (group II) or sildenafil + rosuvastatin (group III) results in a statistically very highly significant (*p* < 0.001) reduction in the Fulton Index, while the difference between groups II and III are not statistically significant; therefore, the addition of rosuvastatin does not provide any additional benefit compared with just sildenafil. However, the same cannot be said in the case of magnesium sulfate. It has an essential role in reducing the RVMI and therefore the degree of RV hypertrophy in the CHPH rat model.

No statistically significant results were recorded regarding the other evaluated parameters (heart weight and body mass).

### 3.3. Echocardiographic Measurements after Treatments

In the statistical analysis of RV anterior wall thickness at diastole (RVAWd), statistically highly significant differences were observed between groups I and III, groups I and IV, groups I and V, groups II and III, groups II and IV, groups II and V, groups III and IV, groups III and V, and groups IV and V. The most prominent RV anterior wall hypertrophy emerged in group I, as shown in Figure 3. Once drug therapy starts, this parameter tends to become progressively reduced: very highly significant (*p* < 0.001) for the combination of sildenafil, 25 mg/kg, i.g. + rosuvastatin, 10 mg/kg, i.g. (group III) or sildenafil, 25 mg/kg, i.g. + magnesium sulfate 10% solution, 10 mL/kg, i.g. (group IV) compared with group II (treated only with sildenafil, 25 mg/kg, i.g.). There is also a very highly significant difference between groups IV and V; therefore, the combination of sildenafil + magnesium sulfate is clearly inferior to the three-drug combination in group V. Of note is that both magnesium sulfate and rosuvastatin in combination with sildenafil reduced the degree of hypertrophy of the anterior wall of the RV, with values approaching those of group C.

In the statistical analysis of the RV internal diameter at diastole (RVIDd) values, statistically highly significant differences were observed between groups I and II, groups I and III, groups I and IV, groups I and V, groups II and III, groups II and IV, groups II and V, groups III and IV, groups III and V, and groups IV and V. Figure 4 shows that the highest RV diameter value appears in group I. However, with the initiation of drug therapy, this parameter shows a statistically very highly significant downward trend (*p* < 0.001). The dual combination of sildenafil, 25 mg/kg, i.g. + magnesium sulfate 10% solution, 10 mL/kg, i.g (group IV) offers very highly significant benefits compared with group III (sildenafil, 25 mg/kg, i.g. + rosuvastatin, 10 mg/kg, i.g). The difference between groups IV and V is also very highly significant; therefore, the combination of sildenafil + magnesium sulfate is clearly inferior to the three-drug combination in group V. However, the three-drug treatment (sildenafil, 25 mg/kg, i.g. + rosuvastatin, 10 mg/kg, i.g. + magnesium sulfate 10% solution, 10 mL/kg, i.g.) manages to reduce the degree of RV dilation in a statistically very highly significant manner (*p* < 0.001), with values approaching those of group C.

In the statistical analysis of the PAAT/PAET ratio, statistically highly significant differences were observed between groups I and II, groups I and III, groups I and IV, groups I and V, groups II and IV, groups II and V, groups III and IV, and groups III and V. The lowest value of this ratio appears in group I, which suggests that the experimental model exposed to 10% hypoxia for 4 weeks developed PH with 77% sensitivity and 80% specificity, showing a ratio < 0.25. With the initiation of drug therapy, Figure 5 shows an upward trend of this ratio following the introduction of sildenafil (group II) or sildenafil + rosuvastatin (group III), with statistically very high significance compared with group I (*p* < 0.001). The highest PAAT/PAAT values can be observed in groups IV and V, suggesting that magnesium sulfate + sildenafil (group IV) has significantly superior effects compared with rosuvastatin + sildenafil (group III) (*p* < 0.001). However, no statistically significant differences were recorded between groups IV and V. Therefore, we can conclude that magnesium sulfate treatment has the most important role in this context.

In the statistical analysis of the PA diameter, statistically very highly significant differences were observed between groups I and II, groups I and III, groups I and IV, groups I and V, groups II and III, and groups II and V. Whether monotherapy, the two-drug combination or the three-drug therapy were administered, PA diameter showed a significant increase, suggesting that the process of PA dilatation is continuous and irreversible, regardless of the associated drug treatment.

### 3.4. Biochemical Parameters

In the statistical analysis of BNP values in RV and PA tissue samples, statistically very highly significant differences were observed between group I subjected to 10% hypoxia (which recorded the highest BNP value) and groups IV and V, as can be seen in Figure 6 and Figure 7. In terms of association with sildenafil treatment (group II), this parameter shows an improving, but statistically insignificant, trend. Group V under the three-drug therapy (sildenafil, 25 mg/kg, i.g. + rosuvastatin, 10 mg/kg, i.g. + magnesium sulfate 10% solution, 10 mL/kg, i.g.) and group IV (sildenafil, 25 mg/kg, i.g. + magnesium sulfate 10% solution, 10 mL/kg, i.g.) have the lowest BNP values. There are no significant differences between group IV and group V. Therefore, in this case as well, the addition of magnesium sulfate has a clear superior benefit in lowering the BNP value. However, the presence of rosuvastatin had a negligible effect in lowering the BNP value in this rat group. No significant differences were observed between those two tissue samples (RV and PA), as we can see in Figure 6 and Figure 7.

In the statistical analysis of VEGF values in both the PA and RV tissue samples, statistically very highly significant differences were observed between groups I and III, groups I and IV, and groups I and V. Group I recorded the highest VEGF value. Statistically very highly significant differences (*p* < 0.001) appeared in groups III (sildenafil + rosuvastatin), IV (sildenafil + magnesium sulfate) and V (sildenafil + rosuvastatin + magnesium sulfate), as shown in Figure 8. Group II, treated only with sildenafil, shows a significant decrease in the VREGF value (*p* < 0.05) compared with group I. However, the lowest VEGF value was recorded in group V, under the three-drug therapy. In this case, both rosuvastatin and magnesium sulfate, but not the combination thereof, showed evident and highly significant benefits in decreasing the VEGF value. No significant differences were observed between those two tissue samples (RV and PA).

In the statistical analysis of NO metabolite values in the PA and RV tissue samples, statistically very highly significant differences were observed between groups I and IV and groups I and V, as shown in Figure 9. The most statistically very highly significant and lowest values (*p* < 0.001) were recorded in group IV under sildenafil + magnesium sulfate therapy and in group V under the three-drug therapy (sildenafil + rosuvastatin + magnesium sulfate). Therefore, the combination of sildenafil with rosuvastatin or magnesium sulfate was observed to have clear benefits (*p* < 0.001) in reducing the NO metabolite values. However, the lowest values of this parameter were recorded in the group under the three-drug combination therapy. No significant differences were observed between those two tissue samples (RV and PA).

No statistically significant differences were observed in the statistical analysis of ET-1 values in PA and RV tissue samples, as shown in Figure 10.

## 4. Discussion

As we know, the literature is still limited with respect to the specific therapy of PH due to lung diseases and/or alveolar hypoxia. This study aims to provide some additional information about the possible combinations of certain classes of drugs studied in this pathology, in order to obtain the best treatment effect. Chronic drug therapy was initiated two weeks after placing the rats in an automatic hypobaric chamber, when, in theory, the process of PH development had already been established [3,4]. The obtained statistical data, with statistically very highly significant results, confirmed that the experimental CHPH model was successful. A significant increase in RV weight was recorded and the Fulton Index, a marker of RV hypertrophy, was increased as well. Two more echocardiographic parameters supported the fact that PH development caused an increase in RVAW thickness and a dilatation of the RVID due to volume and pressure overload in the right side of the heart. An indirect assessment of the increased systolic pressures at the PA level was performed using the PAAT/PAET ratio < 0.25 [6]. According to literature data, a PAAT/PAET ≤ 0.25 or RVAWd ≥ 1.03 mm detected RV systolic pressure ≥ 35.5 mmHg or Fulton index ≥ 0.34 with a sensitivity of 88% and specificity of 100%. The indicated cutoff value of RV anterior wall thickness > 1.03 predicts a Fulton Index > 0.34 with 75% sensitivity and 76% specificity. [6]. Also, CHPH is rarely associated with severe elevation of pulmonary artery pressures [3]. All these processes suggest irreversible PA dilatation. Regarding biochemical parameters, the highest values of the BNP, VEFG and NO metabolites were recorded in group I, exposed only to hypoxia. This suggests that PH development causes myocardial stress due to increased RV pressures [9].

In vivo studies have shown that PVR tended to decrease after 3 weeks of treatment with sildenafil *±* dapagliflozin. This PDE5 inhibitor induces an efficient vasodilatory and anti-remodeling effect on the pulmonary vascular bed in monocrotaline-treated animals [27]. This phenomenon has also been described in CHPH in rats: Sebkhi et al. showed that PDE5 is found in smooth muscle fibers in the vascular tree and that PDE5 inhibition can reduce PA pressure and vascular remodeling when administered ongoing CHPH [28].

Moreover, in this study, sildenafil monotherapy had nevertheless a modest effect on the Fulton Index, as well as on the echocardiographic parameters assessing the hypertrophy degree (RVAWd and RVIDd). However, group II showed a very highly significant decrease in the indirect markers of RV systolic pressure (PAAT/PAET ratio) compared with group C. Sildenafil monotherapy did not significantly reduce the value of the tested biomarkers. Therefore, it can be said that sildenafil provides a modest therapeutic effect by inhibiting the progression of RV hypertrophy and RV systolic pressure.

Rosuvastatin is a hydrophilic statin with important pleiotropic effects that has demonstrated an in vivo benefit in remodeling the small pulmonary artery, normalizing RV hypertrophy and improving RV pressure in both CHPH and MCT-induced PH [17,29]. Beneficial effects mainly appear at the cellular level, such as a pleiotropic effect on endothelial function and NO production.

In this in vivo study, the combination of rosuvastatin + sildenafil showed a significant reduction in RVAWd thickening and RVIDd compared with sildenafil therapy alone. However, it did not provide additional benefits to the Fulton Index or PAAT/PAET ratio compared with sildenafil. This combination also failed to significantly reduce the value of the tested biomarkers. There were only modest decreases in the BNP, VEGF and NO metabolites. Therefore, the additive benefit of the sildenafil + rosuvastatin combination is significant in improving RV hypertrophy.

The dual combination of sildenafil + magnesium sulfate used in group IV showed a statistically very highly significant improvement in the ultrasonographic parameters of RV hypertrophy: RVAWd, RVIDd and PAAT/PAET ratio compared with sildenafil monotherapy or the sildenafil + rosuvastatin combination. Both BNP and VEGF, but especially iNOS, values were very low compared with groups I, II (sildenafil) and III (sildenafil + rosuvastatin).

Magnesium sulfate is an essential element for the functionality of the pulmonary vascular endothelium thanks to its anti-inflammatory properties, and its ability to increase NO production results in important vasodilatory effects [18,25]. Furthermore, experimental studies have demonstrated the beneficial role of this intracellular cation. A diet supplemented with magnesium improved RVMI and PVR and ameliorated PH severity and arterial wall thickening in MCT and CH-treated rats [24].

In this study, the three-drug combination sildenafil + rosuvastatin + magnesium sulfate led to the most highly significant results, with the lowest values of all investigated parameters compared with previous therapies. The Fulton Index improved significantly. The same can be said about the echocardiographic parameters: the lowest values of RVAWd and RVIDd were recorded in group V, close to the values of group C. The degree of RV hypertrophy improved substantially. PAAT/PAAT improved significantly as well but without reaching the values in group C. Even if a decrease in RV systolic pressure was recorded, the three-drug combination merely improves the severity of this parameter.

Concerning biochemical parameters, BNP has the lowest values in groups IV and V. Therefore, magnesium sulfate and rosuvastatin have an essential role in reducing this myocardial stress marker at the RV level. The important vasodilator effects from increasing NO production could explain this fact.

The lowest VEGF value is also found in group V. As we well know, VEGF signaling is involved in pathogenesis and vascular remodeling in PH. VEGF and VEGF receptor 2 are overexpressed in the plexiform lesions of patients with PH [30]. This three-drug combination also appears to have beneficial effects on vascular remodeling when inhibiting this factor.

Nitric oxide (NO) is a free radical involved in multiple biochemical processes. It is synthesized from L-arginine via inducible NO synthase (iNOS). The iNOS isoform is a key mediator of immune activation and inflammation, implicated in the pathogenesis and progression of multiple pathologies, implicitly in PH. iNOS hyperactivation will stimulate NO production at a measure of 1000-fold or greater. This accelerated synthesis has the opposite effect because NO can interact with a superoxide anion to produce NO metabolites involved in tissue injury. Nitrite and nitrate, known as NO metabolites, modulate blood pressure, vascular tone and vascular relaxation [31,32,33].

No statistically significant differences were observed in the statistical analysis of ET-1 values in this study. In the statistical analysis of ET-1 values, no highly statistically significant differences were observed in the AP sample. ET-1 values were maintained at low concentrations between 33 and 50 pg/mL (reference values 7–500 pg/mL). Leary PJ et al., in a recent study, describe the ET-1 paradox; there is a possibility that low levels of ET-1 may lead to heart failure and higher levels of ET-1 may be cardioprotective, and animal models support this. Endothelin receptor antagonism leads to a negative inotropic effect in animal models, which is consistent with other observations that low ET-1 levels have been associated with heart failure [34,35,36].

In this study, the combined use of sildenafil with rosuvastatin or magnesium sulfate brings clear benefits in reducing NO metabolites. However, the lowest value was found in group V.

This study has some limitations: Only one concentration of each drug was administered without the possibility of using multiple concentrations due to the small number of laboratory animals. Another limitation is that the plasma concentration of magnesium was not assayed, especially in groups IV and V. Additionally, the administration of rosuvastatin and magnesium sulfate alone would have provided further valuable information on their impact on the treatment of CHPH. Furthermore, taking tissue samples from the lung and liver, as well as conducting a histopathologic investigation of the pulmonary arteries, lung and liver tissue, could have provided additional information on the adverse effects of the medication and systemic consequences of CHPH.

The results of this study are also preliminary; we might consider that magnesium sulfate and rosuvastatin could be used in conjunction with sildenafil to open a new path in PH group 3 therapy, but the biggest limitation of this study is still the small number of laboratory animals used, so future research is necessary.

However, this three-drug combination could be the basis for new therapeutic strategies in pulmonary hypertension.

## 5. Conclusions

Sildenafil monotherapy does not provide any substantial benefit in improving CHPH in the rat model, but the pleiotropic beneficial effects of rosuvastatin on maintaining vascular endothelial function recommend it as a possible therapeutic class in PH resulting from lung diseases and/or hypoxia. Therefore, under three-drug therapy, the degree of RV hypertrophy (assessed using both the Fulton Index and echocardiographic parameters) and myocardial RV stress marker improved substantially. This three-drug combination resulted in an improvement in pulmonary vascular resistance and afterload. Sildenafil can be used as an anti-remodeling therapy in PH in combination with rosuvastatin and magnesium sulfate. In conclusion, this drug combination could be used to improve the severity of PH and to improve disease prognosis. Treatment should be started in the early stages of the disease. These important findings may underpin new therapeutic strategies for PH treatment in the future. However, future clinical trials on a representative number of patients are still necessary.

## Figures and Tables

**Figure 1 life-14-01193-f001:**
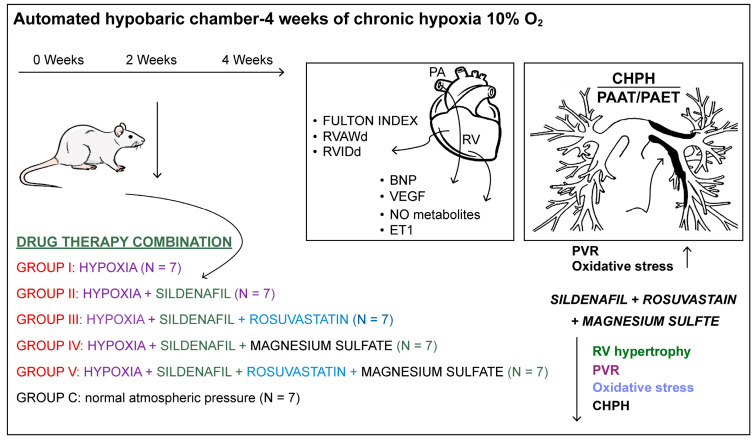
An overview of the CHPH experimental design: 4 weeks of chronic hypoxia, 10% inspired O_2_. Two weeks after exposure, several drugs were administered: sildenafil, rosuvastatin and magnesium sulfate in different combinations. The Fulton Index was calculated as an index for RV hypertrophy. An echocardiographic assessment of cardiac anatomy was performed. RV anterior wall thickness (RVAWd) and RV internal diameter (RVIDd) were measured in diastole; pulmonary artery acceleration time (PAAT), pulmonary artery ejection time (PAET) and PAAT/PAET ratio were calculated. Biochemical parameters (ET-1, BNP, NO metabolites and VEGF) were also calculated.

**Figure 2 life-14-01193-f002:**
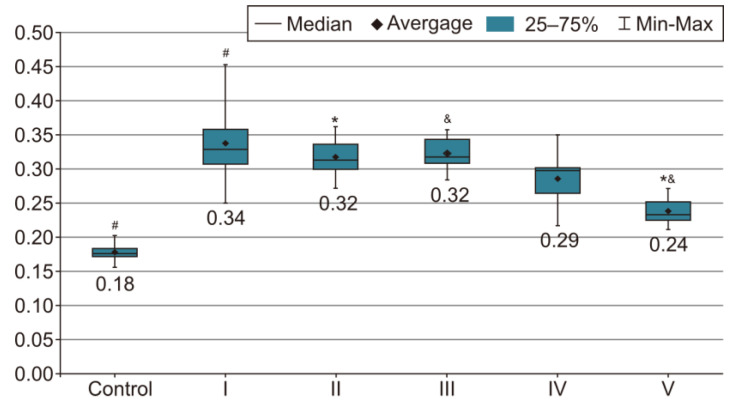
This image presents the Fulton Index, a marker of RV hypertrophy, as the ratio of RV mass to LV mass plus septal mass (RV/LV + S) in rat groups I–V. The results are the means *±* S.D. for 7 animals in each group: group I—hypoxia 10%; group II—sildenafil; group III—sildenafil + rosuvastatin; group IV—sildenafil + magnesium sulfate; and group V—sildenafil + rosuvastatin + magnesium sulfate. # *p* < 0.001, group C vs. group I. * *p* < 0.001, group II vs. group V. & *p* < 0.001, group III vs. group V (which recorded the lowest Fulton Index value compared with groups II (sildenafil) and III (sildenafil + rosuvastatin), with statistically highly significant results (*p* < 0.001)).

**Figure 3 life-14-01193-f003:**
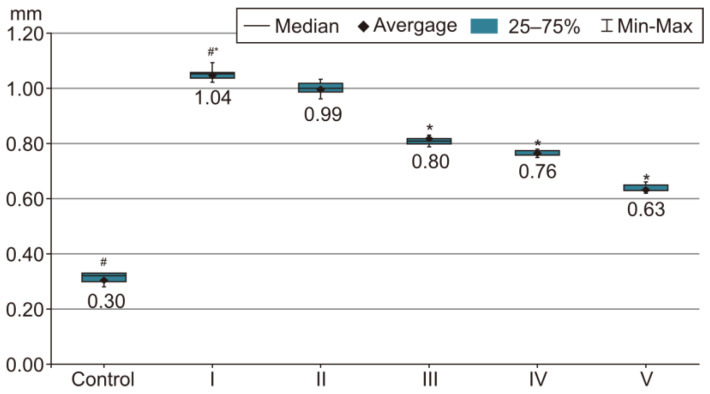
This image presents the RVAWd thickness (mm) and the progressive reduction in this parameter after initiating drug therapy. The results are the means *±* S.D. for 7 animals in each group: group I—hypoxia 10%; group II—sildenafil; group III—sildenafil + rosuvastatin; group IV—sildenafil + magnesium sulfate; and group V—sildenafil + rosuvastatin + magnesium sulfate. # *p* < 0.001, group C vs. group I. * *p* < 0.001, group I vs. groups III, IV and V. The lowest value was recorded in group V (*p* < 0.001) vs. groups II, III and IV.

**Figure 4 life-14-01193-f004:**
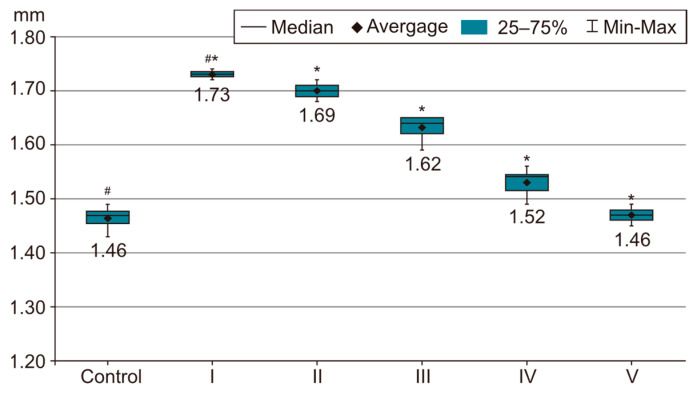
This image highlights how RVIDd (mm) shows a statistically very highly significant decrease with the start of drug therapy. The results are the means *±* S.D. for 7 animals in each group: group I—hypoxia 10%; group II—sildenafil; group III—sildenafil + rosuvastatin; group IV—sildenafil + magnesium sulfate; and group V—sildenafil + rosuvastatin + magnesium sulfate. # *p* < 0.001, group C vs. group I. * *p* < 0.001, group I vs. groups I, II, III, IV and V. The lowest values were recorded in group V (*p* < 0.001) vs. groups I, II, III and IV.

**Figure 5 life-14-01193-f005:**
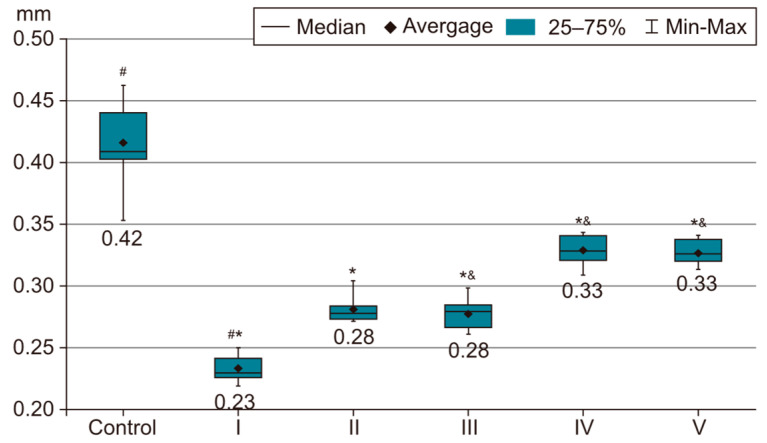
This image shows a graphical representation of the PAAT/PAET ratio in group C and groups I–V under treatment. The results are the means *±* S.D. for 7 animals in each group: group I—hypoxia 10%; group II—sildenafil; group III—sildenafil + rosuvastatin; group IV—sildenafil + magnesium sulfate; and group V—sildenafil + rosuvastatin + magnesium sulfate. # *p* < 0.001, group C vs. group I. * *p* < 0.001, group I vs. groups II, III, IV and V. & *p* < 0.001, group III vs. groups IV and V. The highest values were recorded in groups IV and V (*p* < 0.001) vs. groups I, II and III.

**Figure 6 life-14-01193-f006:**
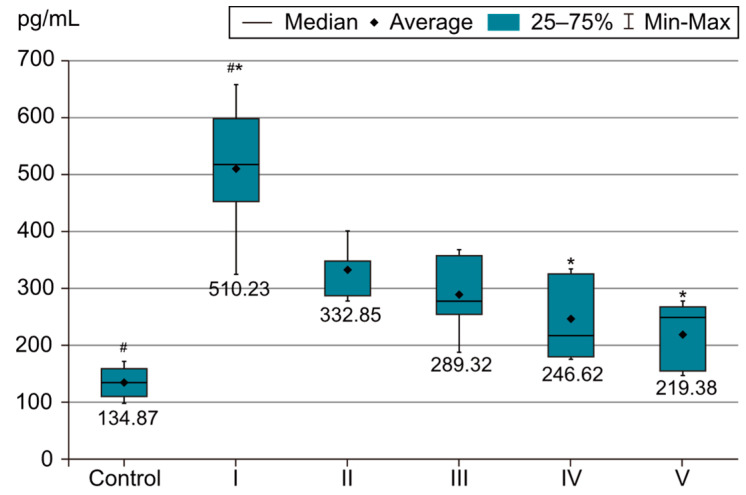
This image highlights the BNP variability in the PA tissue sample (pg/mL). The results are the means ± S.D. for 7 animals in each group: group I—hypoxia 10%; group II—sildenafil; group III—sildenafil + rosuvastatin; group IV—sildenafil + magnesium sulfate; and group V—sildenafil + rosuvastatin + magnesium sulfate. # *p* < 0.001, group C vs. group I. * *p* < 0.001, group I vs. groups IV and V. The lowest values were in group I (*p* < 0.001) vs. groups IV and V.

**Figure 7 life-14-01193-f007:**
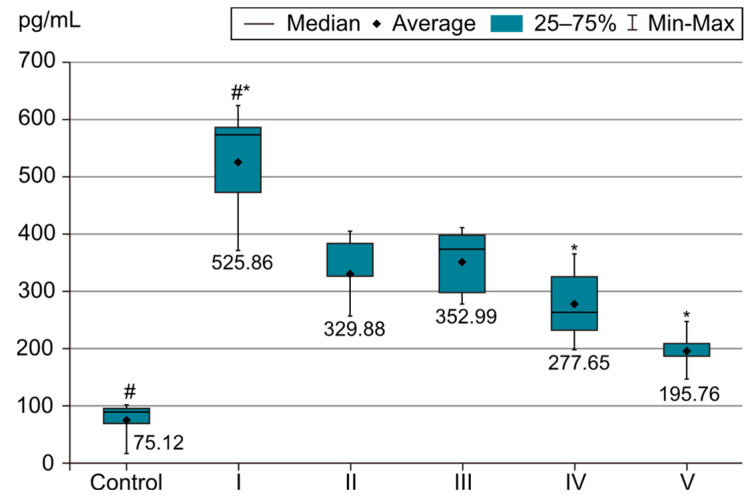
This image highlights the BNP variability in the RV tissue sample (pg/mL). The results are the means ± S.D. for 7 animals in each group: group I—hypoxia 10%; group II—sildenafil; group III—sildenafil + rosuvastatin; group IV—sildenafil + magnesium sulfate; and group V—sildenafil + rosuvastatin + magnesium sulfate. # *p* < 0.001, group C vs. group I. * *p* < 0.001, group I vs. groups IV and V. The lowest values were in group I (*p* < 0.001) vs. groups IV and V.

**Figure 8 life-14-01193-f008:**
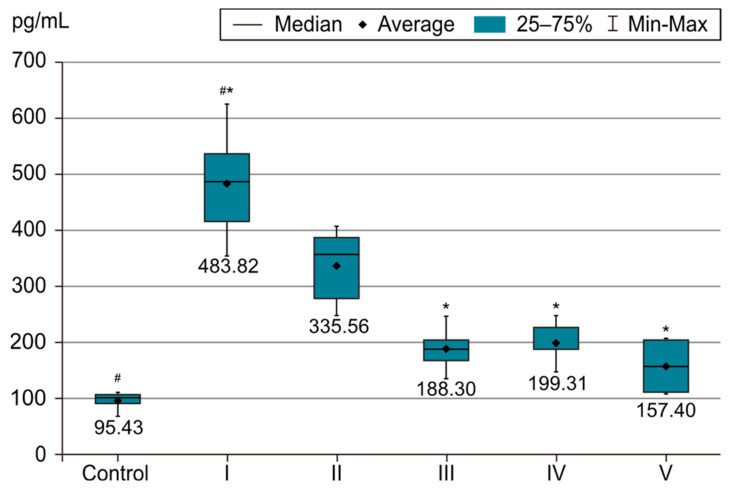
Graphical representation of VEGF in the PA tissue sample. The results are the means ± S.D. for 7 animals in each group: group I—hypoxia 10%; group II—sildenafil; group III—sildenafil + rosuvastatin; group IV—sildenafil + magnesium sulfate; group V—sildenafil + rosuvastatin + magnesium sulfate. # *p* < 0.001, group C vs. group I. * *p* < 0.001, group I vs. groups III, IV and V. The lowest values were in group I (*p* < 0.001) vs. groups III, IV and V.

**Figure 9 life-14-01193-f009:**
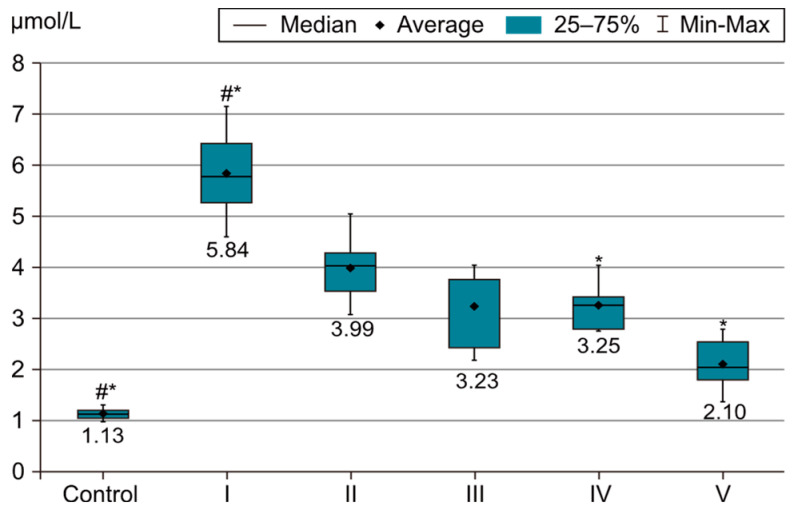
NO metabolites value in PA tissue sample. The results are the means ± S.D. for 7 animals in each group: group I—hypoxia 10%; group II—sildenafil; group III—sildenafil + rosuvastatin; group IV—sildenafil + magnesium sulfate; group V—sildenafil + rosuvastatin + magnesium sulfate. # (*p* < 0.001) Group C vs. group I. * (*p* < 0.001) Group I vs. groups IV and V. The lowest values were in group I (*p* < 0.001) vs. group IV and group V.

**Figure 10 life-14-01193-f010:**
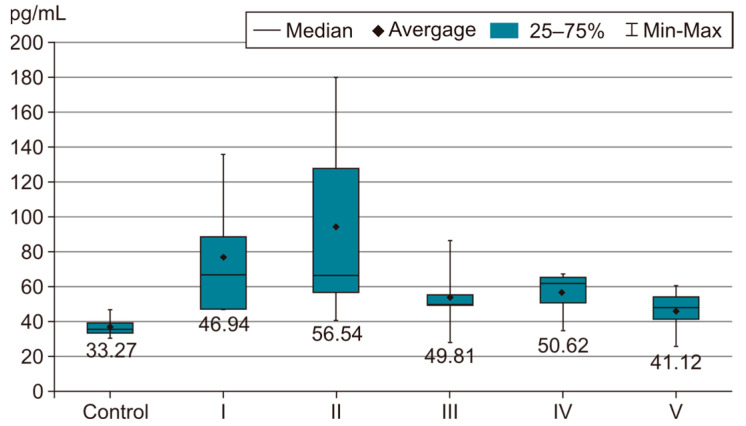
ET-1 value in PA tissue sample. The results are the means ± S.D. for 7 animals in each group: group I—hypoxia 10%; group II—sildenafil; group III—sildenafil + rosuvastatin; group IV—sildenafil + magnesium sulfate; group V—sildenafil + rosuvastatin + magnesium sulfate. No statistically significant differences.

## Data Availability

The data are contained within this article.

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
