# Peer review of "Magnesium Sulfate, Rosuvastatin, Sildenafil and Their Combination in Chronic Hypoxia-Induced Pulmonary Hypertension in Male Rats"

_life, 2024, doi:10.3390/life14091193_

Round 1

Reviewer 1 Report

Comments and Suggestions for Authors

The manuscript is well written and the investigation is interesting and timely. I do not have any changes to recommend. 

Author Response

Response to Reviewer 1 Comments

Comments 1: [The manuscript is well written, and the investigation is interesting and timely. I do not have any changes to recommend]. 

Response 1:  [Thank you for your appreciation.]

Reviewer 2 Report

Comments and Suggestions for Authors

This study evaluated the effect of Mg supplementation on proven therapies of PAH in a rat model. It is concluded that the supplementation of the therapy with Mg is superior against Sil/Rosu therapy.

Major comments:

It is very difficult to follow the authors in this study and understand the protocol. This is based on a combination of very confusing presentation of data on the one hand and the difficulty to understand the study design.

1)      The authors claim that they did everything to reduce the number of rats required for this study but do not explain how they calculated the group sizes. Please add.

2)      In Fig. 1-3 the unit is mm, but this does not make any sense. Please correct and add the correct units.

3)      Although the study is presented in English the authors used in numerical data ‘,’ instead of ‘.’, i.e. ‘0,2’ instead of ‘0.2’. This needs to be corrected.

4)      I could not find data on biomarkers for BNP. I could see data for PA. Did you homogenize tissue or did you use blood samples. For biomarkers you must take blood samples.

5)      I could not see data on PA pressure that are required for this study.

6)      We must know the plasma concentration of Mg for these rats. Do PAH rats have a Mg deficiency? Could you normalize these values? How does Mg improve NO production? Please complete.

7)      In the M&M section the authors report about 7 groups. I could only see data for 6 groups here. A groups receiving only Mg is missing.

8)      If ET is not affected, how did the authors explain the reduction in RV hypertrophy?

9)      How can be the RV hypertrophy so different in the groups but not that of the biomarkers?

10)    Please add the term male in the title.

11)   Why did the rats receive methylcellulose?

Author Response

Response to Reviewer 2 Comments

1. Summary

  Comments 1: [The authors claim that they did everything to reduce the number of rats required for this study but do not explain how they calculated the group sizes. Please add. ….]

Response 1:

 Thank you for pointing this out. We agree with this comment. Therefore, we have added the information regarding this aspect to the manuscript on page 4, lines 6-12.

Animal housing, medication administration, and the entire experiment protocol were performed according to the International Animal Research Guidelines and were approved by the Animal Care and Use Committee of both the University “Iuliu Hațieganu” and the Veterinary Health Authority of Cluj-Napoca, Romania (no. 229/12.08.2020; 260/22.07.2020) and respecting Directive 86/609/EEC.

Council Directive 86/609/EEC of 24 November 1986 on the approximation of laws, regulations, and administrative provisions of the Member States regarding the protection of animals used for experimental and other scientific purposes – recommends a number of 6-8 animals for every individual group.

  Comments 2: [In Fig. 1-3 the unit is mm, but this does not make any sense. Please correct and add the correct units.]

Response 2: Thank you for pointing this out. We agree with this comment. Therefore, we have modified the figures 1-3 in this manuscript.

Because we inserted an abstract Study Design Graph, Figure 1 has become Figure 2 in this manuscript. Since it is a ratio of RV/LV+S, there is no ˝mm˝ unit of measurement. Therefore, we have corrected the figure in the manuscript.

However, there was a technical error in the old Figure 2 (now Figure 3), which we corrected. According to information provided by (Spyropoulos et all. Echocardiographic markers of pulmonary hemodynamics and right ventricular hypertrophy in rat models of pulmonary hypertension. Pulm Circ. 2020 May 29;10(2):2045894020910976) dimensions of right ventricle anterior wall (RVAW) and right ventricle internal diameter (VRID) in diastole, can be measured in the ˝mm˝ unit of measurement. The RVAWd values of healthy adult subjects are 0.32±0.08 mm (as can be seen in the figure below). In this study, the Control group showed a mean value of 0.3 mm. After exposure to hypoxia, a significant increase in this ratio can be observed (in Figure 3), up to a value of 1.04mmm, with a tendency to reduce it under treatment. We believe figure 3 (now Figure 4) does not need to be modified regarding the measurement unit

Spyropoulos F, Vitali SH, Touma M, Rose CD, Petty CR, Levy P, Kourembanas S, Christou H. Echocardiographic markers of pulmonary hemodynamics and right ventricular hypertrophy in rat models of pulmonary hypertension. Pulm Circ. 2020 May 29;10(2):2045894020910976. doi: 10.1177/2045894020910976. PMID: 32537128; PMCID: PMC7268140.     

   Comments 3: [Although the study is presented in English the authors used in numerical data ‘,’ instead of ‘.’, i.e. ‘0,2’ instead of ‘0.2’. This needs to be corrected.]

  Response 3: Thank you for pointing this out. We agree with this comment. Therefore, we have accordingly, modified the numeric data in all figures and manuscript text.

Comments 4: [I could not find data on biomarkers for BNP. I could see data for PA. Did you homogenize tissue or did you use blood samples. For biomarkers you must take blood samples.]

Response 4: Thank you for pointing this out. We agree with this comment. Therefore, we have not used blood samples. 

This article is part of a more comprehensive research study that focuses on the study of tissues.

Indeed, the term ˝ biomarkers˝ is incorrectly used in this context. We replaced it with ˝ biochemical parameters˝ marked in red in the manuscript.

We used a special Elisa kit dedicated to rats, which can be used on several samples: blood, plasma, cell culture, or homogenized tissue culture (Sigma-Aldrich, Romania). BNP is a marker of myocardial stress; tissue was harvested from RV and PA and homogenized.

We have added Fig.7, additional to show that there are no differences between the two samples.

Figure 7. This image highlights the BNP variability in the RV tissue sample (pg/mL). The results are the means ± S.D. for 7 animals each group: group I—hypoxia 10%; group II—sildenafil; group III—sildenafil + rosuvastatin; group IV—sildenafil + magnesium sulfate; and group V—sildenafil + rosuvastatin + magnesium sulfate. # p < 0.001, group C vs. group I. * p < 0.001, group I vs. groups IV and V. The lowest values were in group I (p < 0.001) vs. groups IV and V.

Comments 5: [I could not see data on PA pressure that are required for this study.]

Response 5: Thank you for pointing this out. We agree with this comment. Pressure measurement in PA was not the aim of this study. Recent studies have shown that right ventricular systolic right ventricular pressure ≥35.5 mmHg and Fulton index ≥0.34 are highly sensitive (>94%) and specific (>91%) cut-offs to distinguish animals with Pulmonary Hypertension from controls. When pulmonary arterial acceleration time/ex ejection time (PAAT/PAET) and right ventricular wall thickness in diastole were both measured, a result of either PAAT/PAET ≤0.25 or right ventricular wall thickness in diastole (RVAWd) ≥1.03 mm detected right ventricular systolic pressure ≥35.5 mmHg or Fulton index ≥0.34 with a sensitivity of 88% and specificity of 100% (From Spyropoulos et all. Echocardiographic markers of pulmonary hemodynamics and right ventricular hypertrophy in rat models of pulmonary hypertension. Pulm Circ. 2020).Hypoxic form of PH is rarely associated with severe increased pulmonary artery pressures, according to (Wu, X.; Ma, J.; Ding, D.; Ma, Y.; Wei, Y.; Jing, Z. Experimental Animal Models of Pulmonary Hypertension: Development and Challenges. Anim Models and Exp Med 2022, 5, 211, doi:10.1002/ame2.12220)

We complete the manuscript information on page 12, lines 21-24.

Comments 6: [We must know the plasma concentration of Mg for these rats. Do PAH rats have a Mg deficiency? Could you normalize these values? How does Mg improve NO production? Please complete.]

Response 6: Thank you for pointing this out. We agree with this comment. Therefore, we have used laboratory animals raised and fed standard diets according to international recommendations in this study, which are well-balanced in biological parameters and without nutritional or electrolyte deficiencies. For this reason, we did not test plasma Mg concentration.

We added information about the mechanism by which Magnesium stimulates NO release to the test manuscript, page 3, lines 14-24, and also, above.

(Magnesium, the most abundant intracellular cation, is involved in multiple cellular processes and is an essential element for normal vasoreactivity of the pulmonary endothelium. Increasing Magnesium concentration improves vasodilatation by releasing vasodilator elements that exhibit anti-inflammatory and antioxidant properties. Magnesium sulfate is a natural calcium antagonist. It can upregulate intracellular Ca2+ influx through its mimetic Ca2+ channel-blocking effect, stimulates the production of vasodilator prostacyclin and nitric oxide, and modifies vascular responses to vasoactive agonists. These biochemical reactions control vascular contraction, dilation, growth, apoptosis, and inflammation. Magnesium also suppresses the inhibitory activity of circulating Na+K+ATP ase, which reduces vascular tone. By activating NO synthase (NOS), Magnesium increases the release of NO endothelium, a highly reactive molecule involved in various complex physiological processes. NO is a crucial modulator of vascular homeostasis, maintains vascular tone, and has antiplatelet and antithrombotic properties. Endothelial cells release a constitutive form of NO synthase (eNOS), which generates NO under physiologic conditions. High levels of extracellular Mg stimulate NO production by increasing eNOS levels, whereas no data are available on the effects of low Mg on NO synthesis in human endothelial cells.

Dysfunctions in mg transport systems may predispose patients to cardiovascular disease pathologies. Magnesium efflux occurs via Na++-dependent and Na++-independent pathways. Magnesium influx is controlled by TRPM7 (transient receptor potential melastatin), a non-selective activator of the Ca2+ channel. Thus, TRPM7 may influence intracellular Magnesium levels through alterations in the efflux and influx. such Magnesium in red blood cells, is a more accurate reflection of total body magnesium stores).

   Comments 7: [In the M&M section the authors report about 7 groups. I could only see data for 6 groups here. A group receiving only Mg is missing.]

Response 7: Thank you for pointing this out. We agree with this comment. Therefore, we have noticed an editorial error in the manuscript (page 4, paragraph 2, lines 6-7), for which we apologize and thank you for your observation. The study was divided into 6 groups (not 7), as seen in the Graphic abstract (figure above).

Control group

Group I - rats exposed to hypoxia only 

 Group II - Hypoxia/Sildenafil

Group III - Hypoxia/Sildenafil + Rosuvastatin

Group IV - Hypoxia/sildenafil + Magnesium sulfate

Group V - Hypoxia/Sildenafil + Rosuvastatin + Magnesium sulfate

In this study, we followed the additive effects of Magnesium Sulfate to Sildenafil.

  Comments 8: [If ET is not affected, how did the authors explain the reduction in RV hypertrophy?]

Response 8: Thank you for pointing this out. We agree with this comment. Therefore, we have seen minor changes in ET1 but without statistical significance. I insert the figure above in my manuscript.

Leary PJ et al. (1), in a recent study, describe the ET1 paradox; there is a possibility that low ET1 levels can lead to heart failure and higher ET1 levels may be cardioprotective; animal models support this. Acutely, endothelin-receptor antagonism results in negative inotropy in animal models, which agrees with other observations that lower ET1 levels were associated with heart failure (2). To evaluate the chronic impacts of ET1 difference, genetically modified mice with variable ET1 expression (20%, 65%, wild-type, and 350%) have been studied. Mice with lower ET1 expression develop a dilated cardiomyopathy and die more quickly than wild-type mice (20% expression- death at 560 days; 65% expression- death at 632 days; wild-type/100% expression- death at 841 days). Mice with over-expression of ET1 (350%) had slightly more cardiac hypertrophy and normal cardiac function and lived 876 days (2). This pre-clinical and clinical research agrees with our observation that lower ET1 levels, observed in both PA and RV tissue, can be associated with heart failure and cardiovascular death compared to higher levels.

Other causal explanations are also possible and must be investigated in future studies.

1.     Leary PJ, Jenny NS, et all. Endothelin-1, cardiac morphology, and heart failure: the MESA angiogenesis study. J Heart Lung Transplant. 2020 Jan;39(1):45-52. doi: 10.1016/j.healun.2019.07.007. Epub 2019 Aug 10. PMID: 31515065; PMCID: PMC6942224.

2.      Kelso EJ, Geraghty RF, McDermott BJ, Trimble ER, Nicholls DP, Silke B. Mechanical effects of ET-1 in cardiomyocytes isolated from normal and heart-failed rabbits. Mol Cell Biochem. 1996;157:149–155

3.     Hathaway CK, Grant R, Hagaman JR, Hiller S, Li F, Xu L, Chang AS, Madden VJ, Bagnell CR, Rojas M, Kim H-S, Wu B, Zhou B, Smithies O, Kakoki M. Endothelin-1 critically influences cardiac function via superoxide-MMP9 cascade. Proc Natl Acad Sci USA. 2015;112:5141–5146

   Figure 10. ET1 value in PA tissue sample. The results are the means ± S.D. for 7 animals each group: group I—hypoxia 10%; group II—sildenafil; group III—sildenafil + rosuvastatin; group IV—sildenafil + magnesium sulfate; group V—sildenafil + rosuvastatin+magnesium sulfate. No statistically significant differences.

 Comments 9: [How can be the RV hypertrophy so different in the groups but not that of the biomarkers?]

Response 9: Thank you for pointing this out. We agree with this comment. Therefore, we have rechecked the statistical analysis. According to the data presented in the manuscript, the decrease in the degree of RV hypertrophy (using Fulton Index and RVAWd dimension) as well as that of biochemical parameters, showed a progressive and steady decrease, with statistically very high significant results.

  Comments 10: [Please add the term male in the title.]

Response 10: Thank you for pointing this out. We agree with this comment. Therefore, we have made the title modification.

 Comments 11: [Why did the rats receive methylcellulose?]

Response 11: Thank you for pointing this out. We agree with this comment. Therefore, we have used methylcellulose because it is currently authorized as a feed additive for all animal species without a minimum or maximum content. It is also authorized or used as a food additive. It is intended to be used as a technological additive (category: emulsifier, stabilizer, binder, thickener, and gelling agents) in feeding stuff and premixture for all animal species (according to EFSA Panel on Additives and Products or Substances used in Animal Feed (FEEDAP).

Bampidis V, Azimonti G, Bastos ML, Christensen H, Dusemund B, Kos Durjava M, Kouba M, López-Alonso M, López Puente S, Marcon F, Mayo B, Pechová A, Petkova M, Ramos F, Sanz Y, Villa RE, Woutersen R, Bories G, Gropp J, Nebbia C, Innocenti ML, Aquilina G. Safety and efficacy of methyl cellulose for all animal species. EFSA J. 2020 Jul 31;18(7):e06212. doi: 10.2903/j.efsa.2020.6212. PMID: 32760468; PMCID: PMC7393478

4. Response to Comments on the Quality of English Language

Point 1:

Response 1: The MDPI author service helped improve the English language and figures. The Certificate is inserted below.

5. Additional clarifications

For the research article, we have created a Study Design Chart, which is inserted on page 4, paragraph 4, lines 11-19, to help readers better understand the study design.

Reviewer 3 Report

Comments and Suggestions for Authors

In modern medicine, multidrug therapies are currently of great interest to researchers and clinical practitioners. The research presented here is a good example of the authors joining this research trend. The study presented here describes the use of mono- and multidrug therapy with Sildenafil, Rosuvastatin and Magnesium sulfate in pulmonary hypertension (PH) in a rat model of PH.

After carefully reading the manuscript, I conclude that the abstract, introduction, and other chapters cover the issues discussed in an extensive and proper manner. The conclusions presented by the authors are consistent with the evidence and relate to the major research issue.

The weakness of the reviewed work concern as follows:

1) The small number of citations of works from the last 5 years is rather surprising (out of 41 bibliographic items, only 9 have been published after 2019). I recommend authors to update their reference section.

2) A very important aspect of any research method is the indication of its limitations. The authors have outlined the limitations of their research in the introduction section. However, I believe that the authors should present this aspect in the form of a separate chapter and discuss in more detail the impact of limitations on the validity of the data obtained in this research.

3) In addition, I strongly recommend that the authors should discuss the clinical aspects of mono- and multi-drug therapies in the introductory section. This will allow the reader to better understand the purpose and scientific and practical value of the studies presented.

These suggested changes will allow the reader to better understand the manuscript presented for review.

I recommend publication after major revision.

Comments on the Quality of English Language

.

Author Response

Response to Reviewer 3 Comments

1. Summary

Comments 1: [ The small number of citations of works from the last 5 years is rather surprising (out of 41 bibliographic items, only 9 have been published after 2019). I recommend authors to update their reference section.]

Response 1: Thank you for pointing this out. We apologize for this very important detail. We agree with this comment, so we have made all the necessary changes to the bibliography and manuscript text.

The new bibliographic insertions, marked in red in the manuscript, can be seen below.

Reference [5] - page 2, line 9.

Reference [8] - page 2, line 14

Reference [11] - page 2, line 21

Reference [14]. - page 3, line 4.

Reference [15]. - page 3, line 9

Reference [20]. - page 3, line 20

Reference [21, 22]. - page 3, line 25

Reference [23]. - page 3, line 26

Reference [26]. - page 4, line 2

Reference [27]. - page 12, line 5

Reference [29]. - page 12, line 10

Reference [30]. - page 13, line 2

Reference [34, 35, 36]. - page 14, paragraph 1, line 2

Comments 2: [A very important aspect of any research method is the indication of its limitations. The authors have outlined the limitations of their research in the introduction section. However, I believe that the authors should present this aspect in the form of a separate chapter and discuss in more detail the impact of limitations on the validity of the data obtained in this research.]

Response 2: Thank you for pointing this out. We apologize for this very important detail. We agree with this comment, so we have made all the necessary changes, al page 14, lines 10-15.

This study has some limitations. Only one concentration of each drug administered was given, and the pulmonary artery pressure was not measured using invasive methods. The administration of Rosuvastatin in monotherapy or Magnesium Sulfate in monotherapy could have provided new and valuable information on their impact on the treatment of CHPH. Taking tissue samples from the lung and liver, as well as histopathologic investigation of pulmonary arteries, lung, and liver tissue, could have provided additional information on the adverse effects of the medication and systemic consequences of CHPH.

Weighing the limitations of the study, we see that the results of this study are preliminary; we might consider that magnesium sulfate and Rosuvastatin could open a new way in PH group 3 therapy associated with Sildenafil. So, future research is necessary.

Comments 3: [In addition, I strongly recommend that the authors should discuss the clinical aspects of mono- and multi-drug therapies in the introductory section. This will allow the reader to better understand the purpose and scientific and practical value of the studies presented.]

Response 3: [Type your response here and mark your revisions in red]

Thank you for that comment. We agree with this comment. We have therefore introduced the clinical aspects of mono and multi-drug therapies in the Introduction section, page 3, paragraphs 1, 3, 4 and 5, marked in red.

Point 1: Minor editing of English language required.

Response 1: The MDPI author service helped improve the English language and figures. The Certificate is inserted below.

Round 2

Reviewer 2 Report

Comments and Suggestions for Authors

I thank the authors for their revisions. I realized that the authors corrected a couple of mistakes. However, there are still three questions that the authors did not address: First, how did you calculate the group numbers? This must be done as a statistical calculation and not via recommendation. Please provide the parameter by which you calculated the group sizes. Second, without any knowledge about plasma Mg concentration the observation is uncomplete. You added some references about potential effects of Mg but we do not know whether your treatment changes plasma or cellular Mg concentration. Third, there is still no explanation for the drop in ventricular weight without change in marker gene expression. This is requiring an explanation not a recalculation of the statistics.

Author Response

Response to Reviewer 2 Comments

1. Summary

Comments 1: [ First, how did you calculate the group numbers? This must be done as a statistical calculation and not via recommendation. Please provide the parameter by which you calculated the group sizes.]

Response 1:

 Thank you for pointing this out. We agree with this comment. Therefore, we have added the statistical calculation regarding this aspect.

To determine the appropriate sample size for our study, we used the “pwr” R package

(Champely S.2020. _pwr: Basic Functions for Power Analysis_. R package version 1.3-0).

The parameters used for our calculation were:

Cohen’s f effect size =0.6 (large effect size according to Cohen’s guidelines),

k (number of group) =6,

significance level α =0.05 and

desired power 1-β =0.80

Based on these parameters, the estimated sample size per group was approximately n = 6.922.

This means 7 rats per group to achieve the desired power.

Comments 2: [Second, without any knowledge about plasma Mg concentration the observation is uncomplete. You added some references about potential effects of Mg but we do not know whether your treatment changes plasma or cellular Mg concentration.]

Response 2: Thank you for pointing this out. We agree with this comment. Therefore, we have added the information regarding this aspect below.

      Plasma magnesium levels were not assayed in this study, a limitation we have added to the manuscript. However, information from the literature suggests that at doses below 2 g/kg, there should be no toxicity concerns and that passive internal mechanisms regulating (1) endogenous Magnesium levels are involved in eliminating excess. We used a moderate dose, 10 ml 10% Magnesium sulfate solution in this study. Each 1 ml solution contains 0.4 mmol Mg2+ (equivalent to 100 mg magnesium sulfate). Each 10 ml solution contains 4 mmol Mg2+ (equivalent to 1 g Magnesium sulfate). This is a safe, non-fatal dose, and the rats showed no digestive manifestations, suggesting possible Magnesium sulfate poisoning.

      Our study focused on vasodilator effects of Mg. In clinical practice, short-term magnesium supplementation is performed in various pathologies (arrhythmias and eclampsia) regardless of the plasma magnesium values due to its essential vasodilator, anti-inflammatory, and oxidative stress-reducing effects.

      There are other experimental studies, for example Akkoca et al., in Exp Biomed Res 2019 (1,2), that evaluated Magnesium sulfate's protective efficacy in a rat liver ischemia-reperfusion injury model. By dosing oxidative stress parameters and histopathologic evaluations, they objectified that Magnesium sulfate pretreatment moderately decreased liver damage through its anti-inflammatory and antioxidant effects in a rat liver ischemia-reperfusion model. Neither of these authors considered it necessary to measure plasma magnesium levels.

1.     1. López-Baltanás R, Encarnación Rodríguez-Ortiz M, Canalejo A, et al. Magnesium supplementation reduces inflammation in rats with induced chronic kidney disease. Eur J Clin Invest. 2021; 51:e13561. https://doi.org/10.1111/eci.13561

2.     https://echa.europa.eu/registration-dossier/-/registered-dossier/6440/7/3/1

3.     Akkoca, K., Yoldas, H., Sit, M., Karagoz, I., Yildiz, I., Demirhan, A., … Ozer, B. (2019). Effects of magnesium sulphate on liver ischemia/reperfusion injury in a rat model. EXPERIMENTAL BIOMEDICAL RESEARCH2(3), 93–102. https://doi.org/10.30714/j-ebr.2019353194

Comments 3: [Third, there is still no explanation for the drop in ventricular weight without change in marker gene expression. This is requiring an explanation not a recalculation of the statistics."]

Response 3: Thank you for pointing this out. We agree with this comment. Therefore, we have found an explanation from a pathophysiological point of view.

    Pulmonary Hypertension (PH) results in right ventricular (RV) pressure overload, leading to multi-scale adaptations in RV structure and function in response to the increased afterload. Though the term “cor pulmonale” it is commonly used to describe RV dysfunction as a result of chronic lung disease-associated PH. Because of the typically slow increases over time in pulmonary vascular resistance (PVR) in PH, the RV has time to compensate, often resulting in RV hypertrophy without systolic dysfunction. The concept of coupling is particularly important in physiologically describing the continuum of ventricular adaptation in PH: well-adapted RV often have preserved ventriculo-arterial coupling, while maladapted right ventricles have varying degrees of altered ventriculo-arterial coupling. The "coupling" of the ventriculo-arterial system maintains forward cardiac output. Chronic afterload elevation in PH produces compensatory RV wall thickening, RV Hypertrophy. This increase in RV thickness is accompanied by cardiomyocyte hypertrophy, cardiac ECM remodeling, alterations in cellular metabolism, and an increase in RV capillary density in some models (1,2).

     Chronic pulmonary hypertension activates neurohormonal systems, including the sympathetic nervous and renin-angiotensin-aldosterone systems, increasing contractility and hypertrophy. The primary cause of significant pulmonary hypertension is almost always PVR elevation. Increased RV output alone does not usually cause significant PH because the pulmonary vascular bed vasodilates and recruits vessels in response to increased flow.

     Biochemical parameters are disease-associated molecular changes in tissues and plasma that can serve as standardized, reproducible, noninvasive, and objective measures to assist in the diagnosis, assess prognosis, and monitor response to therapy in PH. Laboratory biomarkers serve as sensitive, but not specific, indicators of RV dysfunction and aid in risk stratification, prognosis, and management (3).

     In this study, therapy with magnesium sulfate, rosuvastatin, and sildenafil demonstrated an important vasodilator role in pulmonary circulation by reducing pulmonary vascular resistance, pressure overload, and afterload. Thus, the degree of RV hypertrophy was significantly reduced. Regarding the biological parameters, BNP, the sensitive and specific parameter in assessing RV myocardial stress, decreased substantially after triple-drug therapy due to the improvement of afterload.

1. Singh N, Dorfmüller P, Shlobin OA, Ventetuolo CE. Group 3 Pulmonary Hypertension: From Bench to Bedside. Circ Res. 2022 Apr 29;130(9):1404-1D22. doi: 10.1161/CIRCRESAHA.121.319970. Epub 2022 Apr 28. PMID: 35482836; PMCID: PMC9060386.

2. Bhattacharya PT, Shams P, Ellison MB. Right Ventricular Hypertrophy. [Updated 2024 Mar 16]. In: StatPearls [Internet]. Treasure Island (FL): StatPearls Publishing; 2024 Jan-. Available from: https://www.ncbi.nlm.nih.gov/books/NBK499876/

3. Pradhan NM, Mullin C, Poor HD. Biomarkers and Right Ventricular Dysfunction. Crit Care Clin. 2020 Jan;36(1):141-153. Doi: 10.1016/j.ccc.2019.08.011. Epub 2019 Oct 21. PMID: 31733676; PMCID: PMC9982435.

Reviewer 3 Report

Comments and Suggestions for Authors

Dziękuję autorom za odniesienie się do moich komentarzy. Uważam, że poprawiony rękopis poprawił przejrzystość.  Dlatego polecam ten artykuł do publikacji

Author Response

Comments: Dziękuję autorom za odniesienie się do moich komentarzy. Uważam, że poprawiony rękopis poprawił przejrzystość.  Dlatego polecam ten artykuł do publikacji   Response: Thank you very much for taking the time to review this manuscript. Thank you for your appreciation.